# *Salmonella* and Salmonellosis: An Update on Public Health Implications and Control Strategies

**DOI:** 10.3390/ani13233666

**Published:** 2023-11-27

**Authors:** Ángela Galán-Relaño, Antonio Valero Díaz, Belén Huerta Lorenzo, Lidia Gómez-Gascón, M.ª Ángeles Mena Rodríguez, Elena Carrasco Jiménez, Fernando Pérez Rodríguez, Rafael J. Astorga Márquez

**Affiliations:** 1Animal Health Department, Veterinary Faculty, University of Cordoba, 14014 Cordoba, Spain; agalanr12@gmail.com (Á.G.-R.); sa2hulob@uco.es (B.H.L.); v32gogal@uco.es (L.G.-G.); mariangii53@gmail.com (M.Á.M.R.); sa1asmar@uco.es (R.J.A.M.); 2Zoonotic and Emerging Diseases (ENZOEM), University of Cordoba, 14014 Cordoba, Spain; bt2cajie@uco.es (E.C.J.); b42perof@uco.es (F.P.R.); 3Food Science and Technology Department, Veterinary Faculty, University of Cordoba, 14014 Cordoba, Spain

**Keywords:** *Salmonella*, salmonellosis, animal health, public health, food chain, predictive microbiology, antimicrobial resistance, control strategies, one health

## Abstract

**Simple Summary:**

*Salmonella* is one of the most important zoonotic pathogen agents, causing an estimated 93.8 million cases of gastroenteritis worldwide annually, with 155,000 deaths. Efforts to reduce transmission of Salmonella by food and other routes must be implemented on a global scale. Salmonellosis control strategies are based on two fundamental aspects: (a) the reduction of prevalence levels in animals and (b) protection against infection in humans. Thus, this review will be focused on *Salmonella* and its relationship between animals and public health (one health approach). The aim is to update the status of *Salmonella* in the world, with special reference to its implications on epidemiology and public health, food chain and risk assessment, antimicrobial resistance, and control strategies. We strongly believe that this review is an opportunity to collect significant and relevant information, using an integral approach, on Animal Health, Public Health, and the relationship between the two.

**Abstract:**

Salmonellosis is globally recognized as one of the leading causes of acute human bacterial gastroenteritis resulting from the consumption of animal-derived products, particularly those derived from the poultry and pig industry. *Salmonella* spp. is generally associated with self-limiting gastrointestinal symptoms, lasting between 2 and 7 days, which can vary from mild to severe. The bacteria can also spread in the bloodstream, causing sepsis and requiring effective antimicrobial therapy; however, sepsis rarely occurs. Salmonellosis control strategies are based on two fundamental aspects: (a) the reduction of prevalence levels in animals by means of health, biosecurity, or food strategies and (b) protection against infection in humans. At the food chain level, the prevention of salmonellosis requires a comprehensive approach at farm, manufacturing, distribution, and consumer levels. Proper handling of food, avoiding cross-contamination, and thorough cooking can reduce the risk and ensure the safety of food. Efforts to reduce transmission of *Salmonella* by food and other routes must be implemented using a One Health approach. Therefore, in this review we provide an update on *Salmonella*, one of the main zoonotic pathogens, emphasizing its relationship with animal and public health. We carry out a review on different topics about *Salmonella* and salmonellosis, with a special emphasis on epidemiology and public health, microbial behavior along the food chain, predictive microbiology principles, antimicrobial resistance, and control strategies.

## 1. Introduction

*Salmonella* spp. is recognized as a major zoonotic foodborne pathogen of economic significance in animals and humans; it causes an estimated 90 million cases of gastroenteritis worldwide annually, with approximately 155,000 deaths [1]. Even though salmonellosis is mostly reported as a foodborne disease, it has been estimated that about 10% of the cases are due to direct contact with animals [2].

*Salmonella* is a genus of highly diverse bacteria that live in the digestive tract of humans and animals. They are widespread in the environment thanks to their ability to survive and adapt even under extreme conditions [3].

Among the over 2600 *Salmonella* serovars described, clinical manifestations and mortality differ depending on both serovar and host characteristics (breed, age, sex, nutrition, and/or immunity) [4].

These serovars are divided into typhoidal and non-typhoidal serovars (NTSs); all of them can cause diseases in animals and/or humans with different levels of severity. Typhoidal serovars are highly adapted to the human host, which is their exclusive reservoir. They are, thserefore, only transmittable through human-to-human contact and cause a potentially life-threatening syndrome known as typhoid (*S. typhi*) or paratyphoid fever (*S. paratyphi*). Most European cases are considered imported cases and generally involve people returning from endemic countries [4].

NTSs are known as zoonotic agents. They are spread from animals and foods to humans but also through human-to-human close contact; they are widely present in the environment and can infect animals and contaminate both water and food. Usually, zoonotic salmonellosis occurs because of a true foodborne infection of animal or plant origin or through close contact with carrier animals [4].

Regarding human infections, only a few of the NTSs are responsible for most human cases. Of these, *S. enteritidis* and *S. typhimurium* are considered the most important serovars with the greatest impact on public health, being responsible for more than 70% of human infections, as illustrated in Table 1.

Most serovars are non-pathogenic for animals but highly pathogenic for humans. Currently, in the European Union (EU), *Salmonella enteritidis* as well as *S. typhimurium* and its monophasic variant are the main serovars responsible for human disease (Table 1). However, *S*. *infantis* serovar has emerged as the fourth most prevalent serovar associated with human disease [4].

The European Food Safety Authority (EFSA) has recently reported an increase in the frequency and severity of human infections caused by *S. typhimurium* and its monophasic variant, which are associated with meat products derived from swine or cattle [5,6,7].

A wide range of domestic and wild animals can host *Salmonella* and thereby become reservoirs: poultry, swine, cattle, wild birds, rodents, pets, and exotic animals. In fact, pets such as dogs and cats and exotic animals such as reptiles and amphibians may play a significant role in transmitting the pathogen in the environment and household by excreting *Salmonella*, which can infect other animals and humans throughout the environment [8,9]. For example, the increase in the private ownership of reptiles has led to a rise in the number of zoonotic infections [10,11,12]. Additionally, rodents and birds act as *Salmonella* amplifiers, playing an essential part in the dissemination of bacteria on farms [13].

The slaughter process can also constitute a source of contamination, especially when adequate hygiene conditions are not maintained. *Salmonella*, which can be present in the intestinal contents of carrier animals, may cause contamination at various stages of the slaughter process, such as in trucks, lairages, slaughter lines, and quartering [14].

The primary clinical manifestation of salmonellosis in humans is self-limiting gastroenteritis, characterized by diarrhea, abdominal pain, fever, headache, nausea, and/or vomiting, which typically resolve within 2 to 7 days. However, in certain cases, especially among children and elderly patients, the illness can progress to a severe and life-threatening condition, accompanied by systemic bacteremia [1].

In contrast, subclinical infections are common in animals, where the bacteria can easily spread between flocks without detection, and animals may become intermittent or persistent carriers. Animals can become infected through different means: (i) close contact with other infected animals; (ii) contaminated water or direct contact with feces due to farm management and/or with contaminated equipment; (iii) transmission from parents to offspring (e.g., *S. abortusovis*, *S. indiana*) [15,16]; (iv) transmission from feed and environment; (v) potential transmission by arthropods.

Regarding foodborne transmission, there is strong evidence from outbreaks in the EU of various foods acting as vehicles [4], e.g., egg and egg products (39 outbreaks), mixed food (24), bakery products (15), pig meat and associated products (14), and vegetables and juices and other products thereof (11). Other food vehicles causing foodborne outbreaks in the EU are raw milk and dairy products made with raw milk, seafood products, and processed foods (e.g., sweets and chocolate) [4].

*Salmonella* strain typing is a crucial component of routine laboratory investigations [17]. Phage typing and serotyping, as well as molecular methods, are essential tools for this purpose. They enable the identification and isolation of such strains from primary animal sources as well as non-animal sources (i.e., food, water, and environmental samples). The most commonly utilized methods include pulsing-field gel electrophoresis (PFGE) and multiple locus variable analysis (MLVA). New genome-based typing methods, such as whole genome sequencing (WGS), are employed to track outbreaks and determine the epidemiological origin of the infection [18,19,20].

Efforts to reduce the transmission of *Salmonella* through food and other routes must be implemented using a one health approach. At the food chain level, the prevention of salmonellosis requires a comprehensive approach at farm, manufacturing, distribution, and consumer levels.

This review provides an overview of the *Salmonella* and salmonellosis status. It is a useful update of concepts for health professionals involved in animal health and public health. It is a review that facilitates and improves the understanding of the epidemiology and control of the main indicator pathogen of zoonoses, *Salmonella*.

## 2. *Salmonella* and Its Relationship with Foodborne Outbreaks: Update in the EU

### 2.1. EFSA Fact Sheet

The European Food Safety Authority (EFSA) provides independent scientific support and advice by collecting and analyzing data on the prevalence of *Salmonella* in animals and foods. It does so by assessing the food safety risks posed by the bacterium for human health and advising on possible control and reduction options. EFSA’s findings are used by risk managers in the EU and the member states in their decision making and support the setting of reduction targets for *Salmonella* in the food chain. EFSA supports the EU’s fight against *Salmonella* using three methods (https://www.efsa.europa.eu/sites/default/files/corporate_publications/files/factsheetsalmonella.pdf, accessed on 15 August 2023) [21]: (i) annual monitoring of *Salmonella* in animals and food to measure its progress; (ii) risk assessments and recommendations; (iii) EU-wide surveys on the prevalence of *Salmonella*.

### 2.2. Foodborne Outbreak Dashboard in the EU

According to Directive 2003/99/EC [22], EU Member States are obliged to report information on foodborne and waterborne outbreaks. The interactive tool (https://www.efsa.europa.eu/en/microstrategy/FBO-dashboard, accessed on 15 August 2023) [23] provides the latest information on foodborne outbreaks and epidemiological information of interest in the year 2021 [4].

The dashboard shows that a total of 4088 foodborne outbreaks (FBOs) occurred in the EU in 2021. From these FBOs, there were 33,813 human cases, with 2560 hospitalizations and 33 deaths. The trends between 2016 and 2021 show a slight decrease in these parameters, with the exception of the death variable, because of the high mortality registered in 2019 and 2020 in human listeriosis cases.

The FBO-dashboard interactive tool shows all known outbreaks and cases per 100,000 people by country, the number of human cases and causative pathogen agents, and the ranking of number of outbreaks by food vehicle and place of exposure.

### 2.3. Salmonella Occurrence in the EU

*Salmonella* spp. is the second most common zoonotic pathogen, after *Campylobacter*, according to the most recent EU One Health Zoonoses Report [4], with both of them causing gastrointestinal infections in humans. The number of confirmed cases of human illness salmonellosis was 60,050, corresponding to an EU notification rate of 15.7 per 100,000 people and with a stable trend between 2017 and 2021. Among these cases, there were 11,790 hospitalizations (45.0% of outbreak-associated hospitalizations) and 71 reported deaths. Furthermore, EFSA data rank it as the leader, causing a total of 773 human cases in foodborne outbreaks (20.8% of outbreak-associated cases), with 1123 hospitalizations and 1 death. *S. enteritidis* was the predominant serovar (N = 350; 79.7% of all *Salmonella* outbreaks). The top five serovars responsible for human infections are currently *S. enteritidis*, *S. typhimurium*, *S. typhimurium* monophasic variant (mST), *S. infantis*, and *S. derby*.

Furthermore, an analysis of the most recent data released by EFSA on the distribution of serovars at the primary-sector level reveals that the majority of *Salmonella* spp. isolates originate from the production of broilers (*Gallus gallus* domesticus) (55.7%), with turkeys (*Meleagris gallopavo*) coming in second with 12.9%, pigs (*Sus scrofa* domestica) with 7.6%, and laying hens (*Gallus gallus* domesticus) with 6.0%. These data were obtained from poultry populations that fall under the purview of the *Salmonella* National Control Program (SNCP) [4]. While *S. infantis* was exclusively associated with broiler sources (95.2%), *S. enteritidis* was mainly associated with broiler flocks and meat (70.0%) and laying flocks and eggs (26.0%). On the other hand, the majority of *S. typhimurium* and mST isolates (43.2% and 65.4%, respectively) were linked to pig sources [4].

A total of 73,238 ‘ready-to-eat’ food sampling units were collected, and they had a very low proportion of *Salmonella*-positive units (0.23%) overall. The highest proportions of positives were found for ‘meat and meat products from pigs’ (0.82%). For ‘non-ready-to-eat’ food, 466,290 sampling units were collected, and the proportion of positive samples was low (2.1%). The food categories with the highest proportions of positive units were ‘meat and meat products’ (2.2%), especially those from broilers (4.4%) and turkeys (3.6%) [4].

A significant increase in the estimated breeding turkey flock prevalence of *Salmonella* was noted in 2021. Flock prevalence trends for target *Salmonella* serovars have, in contrast, been stable over the last few years for all poultry populations.

## 3. An Update on Environmental Stresses Affecting *Salmonella* in Foods

The adaptation of *Salmonella* strains to different environmental stresses in foods has been widely reported, including known increased resistance to low pH, low water activity, and disinfectants, among others. Thus, *Salmonella* remains as an important concern in food processing environments, traditionally linked to the development of greater tolerance and cross-protection mechanisms, thus increasing the persistence along the food chain [24].

Besides the well-known effect of temperature, which is currently applied in pasteurized foods [25], recent studies have shown a growing interest in the acquired resistance mechanisms of *Salmonella* serovars against the presence of acids, low-water-activity foods, and biofilm formation on biotic or abiotic surfaces [26]. All these cumulative hurdles are applied at sub-lethal levels (especially in ready-to-eat foods), so that they promulgate an adaptative response and enable the survival of a larger fraction of *Salmonella* cells.

### 3.1. Acid Resistance of Salmonella in Foods

Acid adaptation allows *Salmonella* to withstand the challenges posed by low pH levels and potentially cause foodborne illnesses. *Salmonella* demonstrates the capacity to modify its physiological characteristics and regulate gene expression against exposure to low pH levels. This adaptation mechanism is especially interesting because of its viability in acidic foods, where conditions might inhibit the growth of other microorganisms [27]. In foodstuffs, weak organic acids like acetic, lactic, and citric acids can be present due to natural food constituents, fermentation processes, or intentional addition during food production to enhance preservation.

The optimum pH for *Salmonella* growth is generally known to be between 6.5 and 7.5, but the minimum pH value depends on many factors, such as the strain, the type of acid, or the synergistic action when combined with other stresses such as NaCl. For instance, Pye et al. [28] compared different *Salmonella* serovars in culture media supplemented with 6% NaCl, 12 mM acetic acid, or 14 mM citric acid, and they found that *S. typhimurium* showed the highest resistance to NaCl and acetic acid stress, while *S. enteritidis* showed the highest resistance level for citric acid.

The increased tolerance to a low pH following acid habituation is referred to as the Acid Tolerance Response (ATR), which has been shown to be strain dependent [29]. This pH-dependent ATR might induce a posterior acid adaptation involving bacteria growth in mildly acidic conditions [30]. These investigations also shed light on the mechanisms of acid-induced cross-protection against ethanol stress in *S. enteritidis* during the growth phase [27], which may lead to more efficient mitigation strategies.

### 3.2. Survival of Salmonella in Low-Water-Activity Foods

The high frequency of *Salmonella* in low-water-activity (a_w_) foods (such as powders, flours, dried fruits, spices, oily foods, and nuts) is a cause for concern. Recent studies have extensively reported on this situation due to the growing number of salmonellosis outbreaks related to these products [31]. These matrices comprise a wide range of the so-called Low-Moisture Foods (LMFs) as being those with reduced water content, making them less favorable for the growth of most microorganisms. According to Food and Drug Administration (FDA) standards, they have an a_w_ at 25 °C of less than 0.85 [32]. These conditions do not allow bacterial growth; however, several studies have demonstrated the survival ability of *Salmonella* spp. in different LMFs [33,34] for months or even years, thus potentially causing adverse health effects for susceptible population groups. Microbial contamination can occur when handling and/or processing any contaminated LMF and/or from environmental contamination, suspended air particles, or inert surfaces [35]. Adaptive responses in *Salmonella* help it survive by accumulating compatible solutes, including proline, glycine, betaine, ectoine, and trehalose, leading to reduced water loss [36]. Furthermore, osmoregulation plays a vital role in maintaining the turgor pressure of the bacteria through increasing the intracellular concentration of compatible solutes [37,38].

Industrial interventions to effectively control *Salmonella* in LMFs have been mainly oriented toward thermal processing. Yet, there is a lack of knowledge on the main factors involved in the thermal resistance of *Salmonella* in LMFs. Liu et al. [39] presented an overview on the factors affecting the microbial safety of LMFs together with the latest developments in analytical methods for the detection of pathogens in dried food commodities. Microbial resistance of *Salmonella* in LMFs can differ according to the type of strain, physiological conditions of the pathogen, food composition (e.g., sugar or fat content), a_w_, and heating temperature [40].

Importantly, the use of cocktails may ensure that novel processes can remove the most resistant strains, as reported in previous studies [41,42]. Some of the latest studies deal with the relationship between moisture content, as a better indicator than temperature, and a_w_ in the thermal inactivation of *Salmonella* in LMFs [43], the effect of food structure combined with emerging technologies [44], and the design of novel test cells to better estimate a_w_ in the thermal resistance of *Salmonella* strains [45]. Despite these recent developments, food industries might still be faced with the randomness and variety of environmental factors associated with *Salmonella* contamination in LMFs, combined with its persistence for a long-term storage period and the difficulties of current methods and sampling strategies in its detection.

### 3.3. The Biofilm Formation of Salmonella in Food-Processing Environments

The different survival mechanisms of *Salmonella*, such as the formation of biofilms, are hypothesized as possible factors for the onset of foodborne diseases. There is clear evidence of the formation of biofilms by *Salmonella* in foods and in different materials present in food processing environments [46,47,48]. *Salmonella* produces a biofilm matrix that is mainly composed of fimbriae (curli) and cellulose [49]. The ability of *Salmonella* to adhere and form biofilms is influenced by multiple factors, such as the composition of the growth medium, the developmental stage of the cells, the characteristics of the inert material, the contact time, the presence of organic substances, and environmental conditions like temperature and pH [50].

Control strategies against *Salmonella* and microbial biofilms overall have been traditionally based on the use of chemical disinfectants, widely applied in the meat industry [51]. However, their effectiveness may differ depending on the type of surface. Other drawbacks such as the increased bacterial resistance to sub-lethal concentrations of disinfectants and the presence of chemical residues preclude their use as a valid antibiofilm strategy. Antimicrobial resistance and toxicity issues have been associated with the use of antibiotics or nanoparticles. Other control strategies still require the application mode and targeted dose to be optimized, as the use of enzymes and quorum-sensing inhibitors are of dubious efficacy against relevant biofilms [52]. Among the physical treatments, pulsed light and UV-C radiation could inactivate the formation of *Salmonella* biofilms [53]. Gao et al. used a combined pulsed light treatment with sodium hypochlorite at moderate levels (100 ppm for 30 min), and it was found to be effective in deactivating a six-cocktail strain of *Salmonella* spp. However, the induction of sub-lethal cells caused by pulsed light deserves further investigation. UV-C radiation alone has overall limited efficacy in reducing *Salmonella* biofilm cells, but its combined use with organic acids with chemical sanitizers seems to be a promising strategy in industrial facilities [54].

Recent developments in biofilm eradication are based on biocontrol strategies such as the use of bacteriophages. Ashrafudoulla et al. [55] evaluated specific lytic bacteriophages against *S. thompson* biofilms on eggshells, which showed better efficacy when using bacteriophage cocktails. In contrast, temperate *Salmonella* bacteriophages can confer greater virulence and resistance to adverse factors, as shown by *S. typhimurium* biofilms. Therefore, the expression of virulence genes and metabolic pathways of *Salmonella* induced by the presence of bacteriophages deserves to be studied further [56]. Another comprehensive study by Asma et al. [57] on natural strategies for biofilm control highlighted the use of plant-based and bee products as antibiofilm molecules. Interestingly, the development of plant-derived nanoparticles (NPs) has arisen as a promising strategy against various bacterial biofilms, including the use of liposomes, cyclodextrins, or hydrogels.

Finally, dual-species biofilms using lactic acid bacteria and/or bacteriocins have been extensively explored as a strategy for the competitive exclusion of *Salmonella* during processing. Research trends are oriented toward the study of extracellular polymeric substances (EPS) to better understand the mechanisms of *Salmonella* biofilm inhibition by LAB and to further explore combinations of LAB biofilms with other LAB metabolites (hydrogen peroxide or bacteriocins) in industrial environments [58].

### 3.4. Predictive Microbiology Models for Estimation of the Microbial Behavior of Salmonella in Foods

Since *Salmonella* can be present in several food commodities, microbial behavior along the food chain has been extensively studied over recent decades. Predictive microbiology is a field that involves using mathematical models and statistical tools to predict the behavior of microorganisms in various environments [59]. Predictive models aim to describe the effect of a certain process (e.g., disinfection, heat treatment, storage, etc.) modulated by a range of environmental factors (e.g., pH, temperature, a_w_, etc.) on the microbial population of interest. Predictions can be quantified through different parameters describing microbial growth, survival, or inactivation, such as maximum growth rate, lag phase, inactivation rate, etc. [60]. As such, applications of predictive microbiology may be oriented to different areas, including food innovation, process control, risk management, reduction of food wastage, design of experiments, and training. There is a wide range of predictive models for describing *Salmonella* behavior in various food categories. Growth, survival, or inactivation ability has been extensively explored in eggs and egg products [61,62,63], meat products [64,65,66], melons [67,68,69], low-moisture foods [40,70,71,72,73], and leafy vegetables [74,75], among others. Furthermore, an extensive review of existing growth/no growth models of *Salmonella* was presented by Carrasco et al. [76] as well as other cross-contamination models [77,78].

While the effect of the most representative environmental factors, such as temperature, pH, and water activity, on *Salmonella* behavior has been properly characterized by the use of dedicated models, research efforts are focused on the effect of emerging preservation technologies or novel antimicrobial agents, as shown in some recent papers. Shahdadi et al. [79] conducted a systematic review and modelling of the role of bacteriophages against *Salmonella* in meat products, while Austrich-Comas et al. [80] evaluated a combined strategy using starter cultures, storage, and high-pressure processing in dry fermented chicken sausages. The use of radio frequency as an inactivation technology against *Salmonella* in treated eggs was successfully modelled by Bermúdez-Aguirre and Niemira [81]. Other models for *Salmonella* using pulsed ohmic heating, UV-radiation, ultrasound, and microwave technologies were reviewed by Alvarenga et al. [82]. It is clear that predictive models can aid in decision making to establish standards for processing by using emerging technologies. Future work should be oriented toward incorporating specific parameters that accurately quantify the effectiveness of emerging technologies in food preservation.

With the advent of dedicated predictive microbiology software, the integration of computational elements is crucial to providing an applicability dimension to predictive models. Machine Learning (ML) algorithms enable computers to learn from and make predictions or decisions based on data. ML models can be trained to identify *Salmonella* genes relevant to disease outcome, thus facilitating the integration of genomic data in microbial risk assessment [83]. Other applications of ML techniques are related to pathogen source attribution [84] and gene-based risk assessments [85]. The inclusion of meteorological factors in ML algorithms on *Salmonella*’s infectivity and outbreak scale was recently reported by Karanth et al., 2023 [86]. The integration of these data with well-defined metadata offers the opportunity for ML models to forecast future trends in antibiotic resistance, determine the sources of pathogens, aid in the investigation of foodborne outbreaks, and enhance risk assessment protocols.

The routine and successful use of mathematical models by the food industry as well as governmental or educational agencies will depend on the development of appropriate and useful applications (software tools) with easy management. There has been an effort to harmonize data formats and model annotations to increase transparency and fit-for-purpose use of predictive microbiology models in a real system. Recent software developments were reviewed by Possas et al. [87]. The authors highlighted the novel fitting shiny apps and improved algorithms that provide better data visualization and graphical representation.

For industrial and health authorities, the use of web interfaces is becoming crucial for a more effective interpretation of predictive models. MicroHibro software (www.microhibro.com, accessed on 15 August 2023) [88] was developed by the University of Córdoba and includes a range of freely available applications related to predictive modelling (safety and shelf life), sampling plans, and risk assessment tools [89]. Currently, MicroHibro software is being updated to include quality or shelf-life models and more advanced risk assessment features.

To illustrate the different predictive model applications for estimating *Salmonella* behavior in the animal-derived foodstuffs supply chain, we show two examples using validated models of pork meat and egg yolk.

MicroHibro contains 27 primary and secondary predictive models of *Salmonella*, including raw vegetable products (tomato, lettuce, avocado, apple, strawberry, cantaloupe), beverages (soya milk), and animal foods (fresh salmon, pork meat, and egg byproducts). Additional models can be included using information from published studies or experimental works. Nevertheless, data curation and inclusion of new models are performed by experts from the University of Cordoba upon request.

The model of Pin et al. [90] was developed for ground pork using data from the literature on different *Salmonella* serovars. The model can describe the microbial fate in the pork supply chain considering that, according to the product formulation, *Salmonella* could tentatively grow or survive during storage. For this case study, growth will be assumed as a function of different pH, a_w_, and temperature conditions. It is well known that before industrial application, predictive models must be validated in the food of interest [59]. Thus, to apply this model to a real processing condition, validation (i.e., comparing predictions of growth responses from the model to actual measures of growth or survival published in the scientific literature) should be performed using observed data for various levels of the environmental factors included within the model domain (Table 2).

In MicroHibro, the user can define validation conditions to assess the closeness of the observed and predicted values. This can be done visually using the equivalence line graph, where predictions are equal to observations, and by comparing the effect of temperature on the µ_max_. In such a way, the validation indices (e.g., bias and accuracy factors) reported by Ross et al. [91] are facilitated in MicroHibro. In Figure 1, the model predictions are relatively close to microbial observations; thus, the model can be used effectively for assessing *Salmonella* growth in the pork supply chain.

Other modelling applications can be used for comparing static and dynamic temperature conditions. As an example, a secondary model for *S. enteritidis* in egg yolk using a dynamic (non-isothermal) profile has been developed [92]. The model can predict the effect of temperature on *S. enteritidis* growth (10–43 °C). Dynamic models can predict the effect of temperature changes over time. Figure 2 shows the comparison between *S. enteritidis* growth in short-term storage (4 h between 15 and 20 °C) and static temperature storage (10 °C). Figure 2 shows that *S. enteritidis* can grow in more than 0.5 log units at dynamic temperatures, thus increasing the probability of a foodborne outbreak through the ingestion of contaminated egg yolk samples (assuming that there are not additional treatments for *Salmonella* inactivation before consumption).

Through these examples, the use of expert computational systems, such as MicroHibro software 3.0, is a powerful tool for supporting food safety and quality activities by Health Authorities and the food industry. This represents a breakthrough in the assessment and management of food safety based on scientific evidence.

## 4. Antimicrobial Resistance

### 4.1. Salmonella and Antimicrobial Resistance: Preface

Several factors of bacterial chromosomes or plasmids may be the cause of *Salmonella* antibiotic resistance [93,94]. These genetic determinants might be in charge of expressing the intrinsic resistance mechanisms linked to the synthesis of beta-lactam antibiotics, alterations in the composition of antimicrobials caused by bacterial enzymes, differences in the permeability of bacteria, the existence of efflux pumps, or changes in target receptors.

The expression of acquired resistance mechanisms, which arise from point mutations in chromosomal genes (e.g., monophasic strains of *S. typhimurium*) or the acquisition of mobile elements like plasmids, transposons, or genomic islands can also result in antimicrobial resistance (AMR) [94]. Resistance transmission can happen vertically between different bacteria or horizontally within the same species or genus. Moreover, it can also occur indirectly through environmental factors [93]. Antimicrobial substances of varying classes, doses, and exposure frequencies are often administered to the gut microbiota of both people and animals for the purposes of treatment, prophylaxis, or metaphylaxis. Additionally, the environment or animal feed sources may contribute to this exposure [95].

A variety of reasons, including improper use of antibiotics in human and veterinary medicine, unhygienic environments and practices in healthcare settings, and pathogens that are resistant to treatment spreading via the food chain, can lead to the development of resistance. Antimicrobials become less effective over time and eventually worthless as a result of this [96]. The primary selective pressure resulting from antibiotic overuse and abuse is still thought to be responsible for the appearance, selection, and spread of microorganisms resistant to antibiotics [97]. Gut bacteria can develop resistance to certain antibiotic substances, which they can then vertically transfer to *Salmonellae* sharing the same ecological niche.

Humans and animals are both impacted by the significant health issue of antibiotic-resistant strains spreading. AMR is still regarded as a zoonosis and poses a major threat to public health, despite efforts in recent decades to decrease the use of antibiotics [98]. Regarding this, a significant problem is the occurrence of Multiple Drug Resistance (MDR) in bacteria, such as *Salmonella* spp., that cause foodborne illnesses that are common around the world. In fact, multidrug resistance (MDR) complicates the use of antibiotics to treat infections and increases the cost of healthcare, lengthens hospital stays, and increases mortality [98].

One of the main causes of global concern for health authorities has been the increase in cases of gastroenteritis and sepsis linked to *Salmonella* strains that are becoming more resistant or even multi-resistant to conventional antimicrobials (e.g., beta-lactams, aminoglycosides, and quinolones, among others). These strains include mainly the Typhimurium serovar and its monophasic variants (mST) [99,100,101] as well as *S. infantis* and *S. kentucky* [102]. Consequentially, resistant infections are on the rise, causing therapeutic failures and longer hospital stays and thus heavily affecting public health and the economy. With over 90,000 salmonellosis cases reported every year in the EU, the EFSA has estimated that the overall economic burden of human salmonellosis could be as high as EUR 3 billion per year (https://www.efsa.europa.eu/sites/default/files/corporate_publications/files/factsheetsalmonella.pdf, accessed on 28 June 2023).

When comparing the presence of *Salmonella* spp. MDR in strains isolated from food and bacteria obtained from animals, relevant research has shown a statistically significant difference. These findings tend to suggest that *Salmonella* strains isolated from food are the main source of MDR [98], which is of great importance when taking into account the fundamental role that this pathogen plays in the food industry and the resistance that has been demonstrated to exist to conventional disinfectants [103]. In light of this, it is crucial to identify the actual animals and people that are the sources of MDR strains in order to reduce their prevalence and enhance public health protection [98].

### 4.2. Key Findings

The annual collection of data on Antimicrobial Resistance (AMR) pertaining to zoonotic and indicator bacteria from humans, animals, and food sources is a collaborative effort undertaken by Member States (MSs) and reporting countries. The resultant datasets are jointly analyzed by the European Food Safety Authority (EFSA) and the European Centre for Disease Prevention and Control (ECDC) and are published in an annual EU Summary Report. The most recent report provides an overview of the key findings of the period from the harmonized AMR monitoring conducted between 2020 and 2021, with a specific focus on *Salmonella* spp. in humans and food-producing animals, including broilers, laying hens and turkeys, fattening pigs, and cattle under 1 year of age along with the associated meat products [104].

Among the reporting countries, the number of *Salmonella* spp. in isolates from human cases varied considerably. Of the 26 reporting countries, including those within the EU and EAA countries, six countries reported very few (<100) human isolates, while three countries reported more than 1000 isolates.

In 2021, overall resistance to ampicillin, sulfonamides, and tetracyclines was noticeably high in *Salmonella* spp. isolates from humans. Similar resistance patterns were observed in isolates from food-producing animals and poultry carcasses, except in the case of laying hens, where resistance levels to these antibiotics were comparatively lower.

Over the period 2013–2021, declining trends in resistance to ampicillin and tetracyclines in isolates from humans was observed in 13 and 11 countries, respectively, coinciding with a decrease in the prevalence of *S. typhimurium*, a serotype commonly associated with pigs and calves. From data reported in 2021, the resistance to fluoroquinolones, specifically ciprofloxacin, was moderate in *Salmonella* isolates from fattening pigs (10.1%) and cattle under 1 year of age (calves) (12.7%). In contrast, in 2020, resistance to ciprofloxacin was noticeably high in isolates recovered from broilers (57.5%), fattening turkeys (65.0%), broiler carcasses (69.3%), and turkey carcasses (46.9%). In 2021, *Salmonella* isolates from humans displayed an average rate of 14.9%, with the lowest levels observed in *S. typhimurium* (7.6%) and *S. typhimurium* monophasic variant (8.9%) and high to extremely high levels in *S. infantis* (33.9%) and *S. kentucky* (78.1%).

It is noteworthy that approximately 95% of isolated *S. infantis* serovars identified in the EU were traced back to broilers and their derived products [102]. Recent research has demonstrated a strong association between the *S. infantis* serovar and elevated antimicrobial and multidrug resistance, resistance to disinfectants, increased tolerance to environmental mercury, heightened virulence, and an enhanced ability to form biofilms and attach to host cells [102].

In contrast, *S. kentucky* isolates from human cases demonstrated consistently high to extremely high resistance levels for ampicillin (62%), ciprofloxacin (77%), tetracycline (57%), sulfamethoxazole (51%), and gentamicin (27.9%). However, resistance to cefotaxime/ceftazidime (6%) and chloramphenicol (12.6%) was observed at low to moderate levels. Similarly, extremely high resistance to ciprofloxacin was reported in *S. kentucky* isolates from broilers (78.0%), laying hens (91.9%), fattening turkeys (96.6%), broiler carcasses (100%), and turkey carcasses (93.3%). In the case of *S. enteritidis*, the most prevalent serovar identified in human cases, resistance to quinolones (ciprofloxacin and nalidixic acid) was 22.6% and 24.8%, respectively [4].

Resistance to third-generation cephalosporins remained notably low in isolates from humans in 2021 (1.1% to ceftazidime and 1.1% to cefotaxime, on average) and was seldom detected in isolates from animal and carcass origins in 2020–2021, except for calves (2.6% to cefotaxime and 1.3% to ceftazidime) and broiler flocks (2.1% to cefotaxime and 2.0% to ceftazidime). Conversely, combined resistance to fluoroquinolones and cephalosporins was very low in isolates from both humans and animals but exhibited higher prevalence in certain *Salmonella* serovars (e.g., *S. kentucky* and *S. infantis*) [4].

Among isolates from human cases, *S. enteritidis* displayed the highest levels of resistance to ciprofloxacin and colistin (22.6% and 17.6%, respectively) when compared to other serovars (14.9% and 5.1%, respectively). Colistin resistance was similarly pronounced in certain *Salmonella* serovars (2020–2021) derived from food-producing animals, particularly laying hens (55.4%) and broilers (53.1%). In these cases, *S. enteritidis* was the predominant serovar [4].

Multidrug resistance was high (22.6%) among *Salmonella* spp. reported in human cases in the EU, ranging from low levels among *S. enteritidis* (1.9%) to very high among *S. kentucky* (54.8%) and extremely high for monophasic *S. typhimurium* 1,4,[5],12:i:- (78.4%). Similarly, MDR was observed at moderate to very high levels in *Salmonella* spp. recovered from carcasses of food-producing animals such as turkeys and broilers (19.1% and 51.2%, respectively) and high levels for all food-producing animals checked, including fattening broilers (41.8%), fattening pigs (39.1%), fattening turkeys (38.2%), and calves (30.4%); the exception was laying hens, which had low-level MDR (6.3%) [4].

The contributions of selected multi-resistant serovars to overall MDR levels in *Salmonella* isolated from animals in 2020–2021 were as follows: broiler carcasses and *S. infantis* (79.4%), broilers and *S. infantis* (73.3%), fattening pigs and Monophasic ST (53.1%), calves and Monophasic ST (41.7%), laying hens and *S. infantis* (34.4%), turkey carcasses and Monophasic ST (24.6%), and fattening turkeys and *S. infantis* (22.1%) [4].

Finally, reporting at the EU level showed that the overall proportion of presumptive ESBL—or AmpC—producers ranged from very low to low among *Salmonella* isolates recovered from all food-producing animal populations and broiler carcasses. It was also very low in isolates from human cases, although higher resistance was observed in specific *Salmonella* serovars (*S. typhimurium* and its monophasic variant and *S. infantis*). No carbapenemase-producing *Salmonella* spp. were isolated from human cases in 2021, nor in animal isolates from 2020 to 2021 [4].

## 5. Control Strategies in Animal Health

Several environmental and management factors have been associated with high levels of *Salmonella* spp. in the animal population. Based on these risk factors, different prevention and control methods related to hygiene and management, health and biosafety, animal welfare, and feeding strategies have been proposed [105,106].

### 5.1. Feeding Strategies

In the case of the poultry and pig industry, the main reservoirs of *Salmonella*, feeding strategies aimed at optimizing intestinal functions may have an impact on the colonization of *Salmonella* in the digestive tract. Among them, we must highlight the acidification of feed by means of organic acids, the use of probiotics, prebiotics, or phytobiotics, and the new lines of research on the incorporation of essential oils (EOs) extracted from plants [107,108]. Most of these products are used in animal health as feed additives, and their approval as therapeutics requires proven scientific studies that demonstrate their antimicrobial efficacy, effect on animal production, and safety for public and environmental health.

The efficacy of EOs obtained mainly from oregano, cinnamon, thyme, and citrus fruits have been evaluated against *Salmonella* serovars [109,110,111]. As an example, the effect of EOs against *Salmonella* serovars isolated from human outbreaks and river water has recently been investigated [108]. This research showed that oregano best inhibited the growth of clinical and environmental Saintpaul, Oranienburg, and Infantis serovars, followed by thyme and grapefruit EOs. The antimicrobial property of the oregano EO, higher than even antibiotic ampicillin, may be attributed to the terpenoids thymol and carvacrol. Therefore, this study concludes that the use of oregano and thyme EOs in conjunction with other oils or bactericidal agents may enhance their effectiveness against infections caused by atypical *Salmonella*. Furthermore, other studies have provided new data on the susceptivity distribution of *Salmonella enterica* strains involved in animal and public health to EOs and a first estimation of the MIC_90_ and MBC_90_ (understood as the Minimum Inhibitory and Bactericidal Concentrations, respectively, able to inhibit or kill 90% of the bacterial population) [109]. The results supported the bactericidal potential of EOs of oregano, common thyme, and red thyme against this bacterium and significant differences between the susceptibility of Typhimurium and Enteritidis serovars. The presence of *S. typhimurium* strains with possible multiple essential oil resistance was also demonstrated.

In addition, different authors have continued their research on assessing the combined effect of these natural substances with traditional antimicrobials (AMBs) as an effective option to reduce bacterial resistance and administration doses [112,113]. In this sense, the synergistic effect between EOs with the main AMBs used against *Salmonella* (enrofloxacin, ceftiofur, and trimethoprim-sulfamethoxazole) has been reported, highlighting the higher percentage of total synergies of trimethoprim-sulfamethoxazole with four EOs (cinnamon, clove, oregano, and red thyme), the most effective combination being enrofloxacin and cinnamon EO [107,113]. These results support the need to expand these trials to more clinical strains and to investigate the mechanisms of action of these synergies.

Based on the above, we believe that the research supports the potential use of EOs (especially oregano, thyme, and cinnamon), alone or in combination, with traditional AMBs, as an effective alternative for the control of *Salmonella* infections of animal or other origin and as a strategy to reduce the development of new bacterial resistance. In addition, we conclude that it is necessary to continue the in vitro studies of susceptibility distribution, the mechanisms that determine the synergy, the in vivo toxicity, and the development of possible resistance mechanisms.

### 5.2. Non-Feeding Strategies

In addition to the feeding-based approach, non-feeding alternatives focus on the use of bacteriophages, vaccines, and the application of biosecurity measures. These strategies are common in poultry and pigs to minimize *Salmonella* prevalence in farms [105,106].

#### 5.2.1. Bacteriophages or Phages

Bacteriophages or phages are viruses that infect and replicate in bacteria until they lyse. They have a capsule and genetic material like eukaryotic viruses. They are natural bactericides and probably one of the most widely distributed microorganisms in the biosphere. Despite their potential usefulness in the treatment of infections, the study of their feasibility has been relegated to the use of antibiotics. In the current context, with the reduction and/or withdrawal of antibiotics from the medical-veterinary scene, alternatives such as phages or bacteriophages may be useful for the treatment and control of bacterial infections such as *Salmonella* [3].

When it comes to prophylaxis, animal therapy, and reducing the number of bacteria in animal-based food products, bacteriophages are thought to be a valuable alternative to antibiotics [114]. Their host-specificity makes them natural, non-toxic, and feasible for therapeutic application, allowing them to attack only the targeted bacteria while safeguarding the rest of the microbiota. Since the immune system can tolerate phages well, they also have the advantage of preventing host allergies [105]. Moreover, they are able to combat resistance to antimicrobial bacteria [115]. *Salmonella* and other foodborne infections have been successfully treated in a number of experiments involving germ-free chickens raised in battery cages [116].

Phage-based methods of controlling *Salmonella* have been tested in poultry [117,118,119,120] and, to a considerably lesser extent, in pigs [121]. In fact, the environment found in chicken farms may be a valuable source of *Salmonella* phages. It has been found that broiler chicken farms in Spain have more diversified *Salmonella* bacteriophages than layer ones based on the most common serovars [118]. However, more research is required to understand the epidemiology of phages in relation to other serovars.

Furthermore, some researchers have recently investigated the use of microencapsulated bacteriophages incorporated into feed for *Salmonella* control in poultry [119,120]. In a first study, in vitro and in vivo gastrointestinal survival of non-encapsulated and microencapsulated *Salmonella* bacteriophages and its implications for bacteriophage therapy in poultry were reported. Significant differences were observed in the results between the phage delivery of in vitro studies compared with in vivo studies [119].

A second study showed that adding the L100 encapsulated phage as a feed additive to the starting diet during rearing could significantly reduce the incidence of flock contamination with *S. enteritidis*. At the conclusion of the rearing period, this pathogen had been fully eradicated from the environment, and there was a decrease in *Salmonella* colonization and excretion. Nevertheless, higher phage doses, better delivery protocols, and/or the combination of different approaches might be required [120].

Finally, other studies have assessed the effect of bacteriophages against *Salmonella* Infantis and *Salmonella enteritidis* on farm surfaces, evaluating bacteriophage application as a complementary tool for cleaning and disinfection procedures [117].

#### 5.2.2. Vaccines

Strategies based on vaccination for the control of *Salmonella* spp. have proven to be a very effective tool for controlling salmonellosis in species such as poultry. For that reason, the manufacturing of vaccines for the poultry industry is based on strains of *S. enteritidis* and *S. typhimurium* [122].

On the contrary, in swine there are currently no effective commercial vaccines. The main problem with medical prophylaxis against *Salmonella* in swine is that there is no cross-immunity between the different serovars (e.g., Typhimurium, Rissen, Derby, Anatum, Bredeney, etc.); therefore, it would be necessary to use specific vaccines (autologous or autovaccines) against the serotype involved in the infection/disease on the farm or to design vaccine candidates that included the predominant serotypes in the geographical area and/or farms involved [3].

The different types of vaccines available on the market are live-attenuated, inactivated, and subunit vaccines [123]. The protection conferred by live vaccines is theoretically greater since they promote a cellular-based response, which *a priori* is ideal for facultative intracellular pathogens such as *Salmonella*. In addition, if they are administered orally, they will manage to produce immunoglobulin-A in the intestine, the main component of the immune system in the control of digestive pathogens. However, these vaccines have certain disadvantages, such as the need to withdraw any antibiotic treatment during oral administration of the vaccine, their cost, and the potential risk of reversion to virulence and biosafety [3]. In fact, secondary mutations in live vaccines can cause reversion to virulence, which affects the overall health of flocks and thus contaminates the environment [124]. To conclude, it is important to note that a vaccine should be safe, give protection against various serovars, and stimulate the host’s immunity system.

Finally, the application of vaccines can have negative effects, such as the development of antibodies (because of vaccination) that interfere with or mask the antibodies developed by the infection. This becomes a problem in countries with a control program based on serological analysis, since the techniques used do not allow for the differentiation of vaccine antibodies from those produced by the infection. There are alternatives to this, such as ELISA techniques that make it possible to differentiate vaccine antibodies from antibodies produced by natural infection (DIVA strategy, Differentiating Infected from Vaccinated); however, these entail additional costs in the surveillance and control of *Salmonella* [3].

#### 5.2.3. Biosecurity

Biosecurity is the most effective and inexpensive disease control measure, aimed at managing the risks posed by diseases to the economy, environment, and human health [125]. The application of biosecurity measures to reduce the levels of prevalence of infections/diseases, with special attention to those that pose a risk to public health (e.g., salmonellosis), should be one of the main objectives of health authorities [126].

The application of strict hygiene and biosecurity measures not only improves the situation of farms with respect to specific pathogens but also improves the overall health of farms. In the specific case of *Salmonella* spp., in intensively reared white pigs and in intensive poultry farming, numerous biosafety protocols and practical guides have been described both at the farm and slaughterhouse level. The most critical points are related to cleaning and disinfection protocols [3].

##### *Salmonella* Cleaning and Disinfection Protocols

The objective of sanitation is to clean and disinfect equipment and materials that enter or remain on farms, including the personal hygiene of farm staff. Following the sanitation program guidelines helps to exclude the presence of pathogens on the farms before they can be spread [126]. As an example, the efficacy of disinfectant misting in the lairage of a pig abattoir to reduce *Salmonella* in pigs prior to slaughter has been reported [127]. This comprises the following: (1) application of high-pressure water to remove organic matter; (2) use of detergent with rinse (e.g., sodium hydroxide or hypochlorite); (3) use of disinfectant without rinsing (e.g., chlorocresol or quaternary ammonium); (4) drying for at least 24–48 h; and (5) fumigation based on cypermethrin.

##### Other Aspects Related to Biosecurity

In this sense, it is necessary to highlight the correct control of rodents on the farm as a basic topic within the *Salmonella* control program. Rodents can carry not only *Salmonella* but also a host of microorganisms: *Campylobacter* spp., *Lawsonia intracellularis*, *Leptospira* spp., *Brucella* spp., *Triquinella spiralis*, and porcine reproductive and respiratory syndrome (PRRS) virus [126].

The control of rodents must be based on good knowledge of their ethology, to locate the refuge points, breeding nests, and passage areas on the farm and to effectively use baits with authorized rodenticide products. All the actions carried out in the rat extermination program must be registered. Likewise, evaluation and verification of the program should be carried out periodically to make modifications if a reduction in the effectiveness of the strategy followed or product used is detected. Finally, it is necessary to indicate that rodenticides must be replaced periodically to avoid tolerances [126].

Wild birds can also act as authentic amplifying reservoirs of different *Salmonella* serotypes. The implication of different *Salmonella* serovars transmitted through birds (e.g., pigeons, turtledoves) acting as the main vectors in disease outbreaks in farms has been reported [16]. Therefore, in livestock farms it is necessary to use anti-bird mesh on windows and access points as well as other preventive biosecurity measures such as closed warehouse doors, permanently closed silo lids, closed feed, and raw material stores, to prevent the access of birds [3].

## 6. Conclusions

Efforts to reduce the transmission of *Salmonella* through food and other routes must be implemented using a One Health approach. The control of salmonellosis is based on two fundamental aspects: the reduction of prevalence levels in animals and the protection of humans from infection.

At the food chain level, the prevention of salmonellosis requires a comprehensive approach at farm, manufacturing, distribution, and consumer levels. Food operators and health authorities play a crucial role in preventing *Salmonella* transmission to consumers by ensuring safe food handling, monitoring and enforcing hygiene standards, and swiftly responding to foodborne outbreaks. Their collaboration safeguards public health and reduces the risk of foodborne illness, underscoring the importance of their roles in safeguarding food safety.

A significant concern is the rise of *Salmonella* MDR strains, which are responsible for foodborne illnesses that are common throughout the world. In fact, multidrug resistance (MDR) complicates the use of antibiotics to treat infections, raises healthcare expenses, lengthens hospital stays, and increases mortality.

Finally, several environmental and management factors have been associated with high levels of *Salmonella* spp. in the animal population. Based on these risk factors, different prevention and control methods related to hygiene and management, health and biosafety, animal welfare, and feeding strategies have been proposed.

## Figures and Tables

**Figure 1 animals-13-03666-f001:**
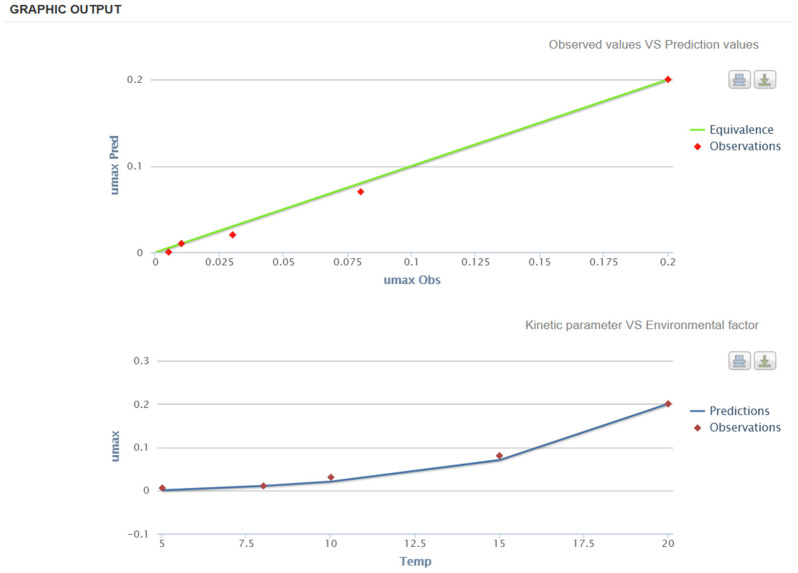
Validation of the model [90] for estimating the growth of *Salmonella* spp. in the pork supply chain using MicroHibro. Top: comparison between observed and predicted maximum growth rates (*y*-axis) (µ_max_, log CFU/h); bottom: evolution of maximum growth rates (µ_max_, log CFU/h) against storage temperature(*x*-axis).

**Figure 2 animals-13-03666-f002:**
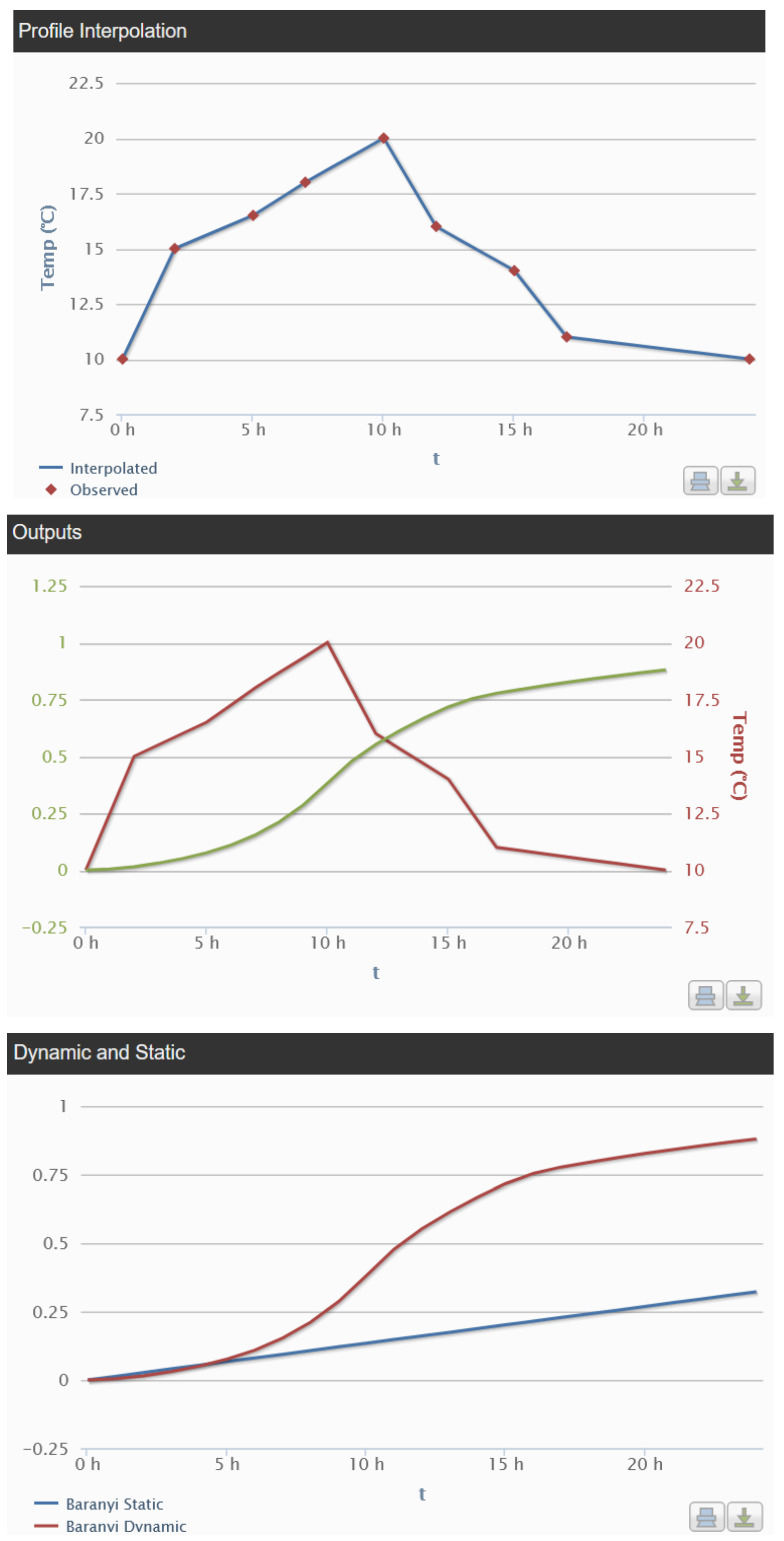
Predictions of *S. enteritidis* growth in egg yolk at dynamic temperatures obtained using the model of Gumudavelli [92] in MicroHibro. Top: representation of the dynamic temperature profile; middle: growth of *S. enteritidis* at dynamic temperatures; bottom: comparison of *S. enteritidis* growth in egg yolk under dynamic conditions and at a static temperature of 10 °C.

**Table 1 animals-13-03666-t001:** Distribution of confirmed cases of human salmonellosis acquired in the EU Member States (MSs), 2019–2021, for the six most frequent *Salmonella* serovars in 2021 [4].

Serovar	2021			2020			2019		
	Cases	MSs	%	Cases	MSs	%	Cases	MSs	%
Enteritidis	23,634	23	64.6	21,203	23	63.1	32,010	24	61.6
Typhimurium	4027	23	11.0	3702	22	11.0	6044	24	11.6
Monophasic Typhimurium 1,4,[5],12:i:-	1269	14	3.5	1530	16	4.6	2668	17	5.2
Infantis	633	23	1.7	716	21	2.1	1215	24	2.3
Derby	239	16	0.7	260	17	0.8	396	20	0.8
Coeln	315	14	0.9	201	17	0.6	270	15	0.5
Other	6462	-	17.7	6009	-	17.9	9378	-	18.0
Total	36,579	23	100	33,621	23	100	52,001	24	100

**Table 2 animals-13-03666-t002:** Dataset used for the model validation of [90] against temperature, pH, and a_w_ conditions and their respective observed maximum growth rates (µ_max_, log CFU/h).

T (°C)	pH	a_w_	µ_max_
5	5.2	0.970	0.005
8	5.7	0.970	0.010
10	5.7	0.976	0.030
15	5.9	0.980	0.080
20	6.0	0.990	0.200

## Data Availability

Not applicable.

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
