# Peer review of "Salmonella and Salmonellosis: An Update on Public Health Implications and Control Strategies"

_animals, 2023, doi:10.3390/ani13233666_

Round 1
Reviewer 1 Report
Comments and Suggestions for Authors
Estimated Authors,
Thank you for your contribution with this review manuscript.
The Astorga Marquez et al., manuscript attempts to review the state-of-the-art on different topics related to Salmonella. In general, the manuscript and the topic are appropriated and in line with the topic of the Journal. However, critical issues should be addressed by the Authors.
The most important issue it should be addressed is the high percentage of similarity detected. Using Turnitin software with exclusion of the bibliographic section and a minimum similarity threshold of 15 words, a substantial similarity with already published works has been identified (total percentage of similarity= 30%). I suggest the authors be careful to avoid this as it may lead to penalties. Below, I have listed some of the compromised text fragments and their corresponding sources:
-L57-62: https://www.mdpi.com/journal/microorganisms/special_issues/Salmonella?cv=1
-L93-101, L158-166: https://www.mdpi.com/2076-2607/11/7/1765?cv=1
-L167-176: https://doi.org/10.2903/j.efsa.2022.7666
-L382-387: https://doi.org/10.3390/microorganisms9102018
-Section 5.2. Key findings. Should be rewritten as is highly similar to what reported in the following reference: https://doi.org/10.2903/j.efsa.2023.7867
-L545-550: https://www.mdpi.com/1660-4601/20/17/6654
-L554-561: https://www.mdpi.com/2076-2615/10/9/1456?cv=1
-L568-573: https://doi.org/10.1016/j.vetmic.2022.109579
-L500-509: https://doi.org/10.1093/lambio/ovad045
Furthermore, although I acknowledge that English is not the authors' native language, I suggest they extensively revise their writing style and, if feasible, employ a more academic language. Additionally, the authors' use of lengthy sentences occasionally makes it challenging to comprehend the intended message, which is unfortunate. Therefore, paraphrasing and replacing terms throughout the text can enhance the arguments' readability and comprehensibility.
The introduction section requires reorganization. The ideas presented are muddled and repetitive. It is recommended to rearrange the main concepts of the section by following a logical sequence. One possible approach would be to commence by elaborating on Salmonella- what it is, the diversity of its genus, i.e. TS and NTS, and its ecology, including the role of various animal species and associated serovars. After explaining that, you can concentrate on the food supply chain, then look at the impacts on animal and human health.
Section 4 ‘Application of predictive microbiology software for Salmonella in foods’ (L282-351) focuses on using predictive modelling software developed by the Authors. However, as a review paper, it may be advisable to examine the current state of research in this area from various perspectives rather than providing a practical illustration and only focus on foods. For example, Machine learning algorithms has been also used for a variety of purposes, such as linking human outbreaks to specific food sources, detecting the occurrence of Salmonella serovars in human infections, and predicting future outbreaks. I encourage you to provide a comprehensive overview of this field and the work that has already been done. Then, if appropriate, you can expose the example already provided.
For all these reasons, the article will be only accepted after major revisions.
Minor comments
L36-38: The cited reference n.3 is not relevant to justify the two previous statements. Please, use only relevant papers.
L44: ‘…the increase in the reptile population…’ What do you mean? Owned reptiles? Please, be more specific.
L45-46: Some rephrasing is needed. Please consider this example: “Additionally, rodents and birds act as Salmonella amplifiers, playing an essential part in the dissemination of bacteria on farms [9].”
L48-50: Also here, need for rephrasing. Here you find an option: “Salmonella, which can be present in carrier animals' intestinal contents, may cause contamination at various stages of the slaughter process, such as in trucks, lairages, slaughter lines, and quartering [10].”
L58: “on both bacterium and host characteristics” the bacterium is always Salmonella; I think you mean here the serovar. Please, replace the term. Instead, what do you mean with host characteristics? In terms of immunity? Please, be more factual.
L79-83: The concept is repeated here from L59-61.
Table 1: from the title ‘reported’ is not needed. Please, delete it. The acronym MSs requires clarification. I believe you intend to refer to Member States, so you could include this term in the table title or as a footnote. Additionally, you have rounded up most of the percentages, unlike those from Derby and Coeln serovars. For uniformity, I recommend rounding up these percentages too.
L89: the first use of EFSA acronym should be expanded as the European Food Safety Authority (EFSA) Consider the following rephrasing: “Recently, the European Food Safety Authority (EFSA) has reported an increase in the frequency and severity of human infections caused by the Typhimurium serovar and its monophasic variant (mST strains), which are associated with meat products derived from porcine or bovine species.”
L122: support and advice for whom? Collecting data from where?
L134: please, move the date of access to the References section
L146: replace ‘puts’ by ‘ranks’
L200: They, who? I suggest modifying the text as follows: “… such as NaCl. For instance, Pye et al. [26] compared…”
L203: ‘while S. Enteritidis showed the highest resistance level for citric acid.’
L207-211: It is a long sentence. Please, rearrange it, otherwise the meaning of the text can be lost.
L223: ‘and/or’
L228: Please start the sentence by stating the author's name instead of just the reference number (e.g., Liu et al. [35]). Additionally, it is advisable to avoid biased language and the use of subjective terms such as 'nice'.
L270: Which research? As mentioned above, please provide the name of the author to make explicit what you are referring to.
L278: again, what do you mean with ‘the latest research’? the ref. [52]?
L283: What do you mean with ‘microbial fate’?
L308-351: This section is a little confusing… The message is sometimes lost. Please, rearrange it.
L334-336: It should be the opposite, I mean the predictions are relatively close to observations but not the contrary. Please, rearrange the sentence.
L341: Figure 3 in this manuscript version does not correspond with you are mentioning. Maybe did you mean Figure 2?
Figure 2. Could you provide the Y-axis titles in the middle and bottom graphs?
L376: ‘Kentucky’. Please correct it.
L379: 90,000 cases of resistant salmonellosis? Please follow a logical order in your arguments. You discussed antimicrobial resistance earlier and now you are presenting data on human salmonellosis cases. This has already been addressed at the beginning of the manuscript, so it is unnecessary to include it again here.
L415-463: References are missing… please provide them.
Figure 3 and 4: I am not certain that you can use the same figures as reported by the cited reference. Can you recreate the figure instead of using the same one from the cited reference? You could declare that the modification was made from [67].
L511: what do you mean with MIC90 and MBC90. It is the first time they appear in the text. Please, explain it.
L515: the acronym ‘MEOR’ is not necessary, as you don’t use it anywhere else in the manuscript. Please, provide only the necessary abbreviations throughout the text.
L633-638: Is this still necessary to explain here? It is repetitive from the Introduction section.
References
The reference section should be revised in the format described in the Journal’s guidelines.
I have detected some errors in ref. n. 5, 8, 10, 11, 58, 62, 68
Ref. n. 20 needs for a title, the url and the date of access
Ref. n. 22 needs for a title of the consulted section of the website and a date of access
Comments on the Quality of English LanguageAlthough I acknowledge that English is not the authors' native language, I suggest they extensively revise their writing style and, if feasible, employ a more academic language. Additionally, the authors' use of lengthy sentences occasionally makes it challenging to comprehend the intended message, which is unfortunate. Therefore, paraphrasing and replacing terms throughout the text can enhance the arguments' readability and comprehensibility.
Author Response
Authors are very grateful for your kind and constructive comments about our original manuscript and your willingness to enrich our work. We hope the questions and comments of the reviewers had been properly addressed and the manuscript will now meet the required standard to be accepted for publication in Animals.
Following, you can find the changes that have been performed, which have been highlighted in the newly revised manuscript:
-L57-62: https://www.mdpi.com/journal/microorganisms/special_issues/Salmonella?cv=1
This information comes from the EFSA (European Food Safety Authority). Given it is a public organization, data is duly referenced and follows the guidelines for authors of Animals in terms of copyright, we do not consider that the paragraph needs to be modified for fear of distorting the official information it contains.
-L93-101, L158-166: https://www.mdpi.com/2076-2607/11/7/1765?cv=1
This information again comes from a public organization (WHO, World Health Organization) and is correctly referenced. It is true that the article the reviewer provides uses this information, but is obtained from the same source and duly referenced. From the same reason expressed above, the paragraph has not been modified. .
-L167-176: https://doi.org/10.2903/j.efsa.2022.7666
This information comes from the EFSA (European Food Safety Authority). Given it is a public organization, data is duly referenced and follows the guidelines for authors of Animals in terms of copyright, we do not consider that the paragraph needs to be modified for fear of distorting the official information it contains.
-L382-387: https://doi.org/10.3390/microorganisms9102018
Following the reviewer’s instructions, this compromised text fragment has been rewritten to avoid similarities with the original works where the information was found.
-Section 5.2. Key findings. Should be rewritten as is highly similar to what reported in the following reference: https://doi.org/10.2903/j.efsa.2023.7867
This information comes from the EFSA (European Food Safety Authority).
-L545-550: https://www.mdpi.com/1660-4601/20/17/6654
Following the reviewer’s instructions, this compromised text fragment has been rewritten to avoid similarities with the original works where the information was found.
-L554-561: https://www.mdpi.com/2076-2615/10/9/1456?cv=1
Following the reviewer’s instructions, this compromised text fragment has been rewritten to avoid similarities with the original works where the information was found.
-L568-573: https://doi.org/10.1016/j.vetmic.2022.109579
Following the reviewer’s instructions, this compromised text fragment has been rewritten to avoid similarities with the original works where the information was found.
-L500-509: https://doi.org/10.1093/lambio/ovad045
Following the reviewer’s instructions, this compromised text fragment has been rewritten to avoid similarities with the original works where the information was found.
- The introduction section requires reorganization. The ideas presented are muddled and repetitive. It is recommended to rearrange the main concepts of the section by following a logical sequence. One possible approach would be to commence by elaborating on Salmonella- what it is, the diversity of its genus, i.e. TS and NTS, and its ecology, including the role of various animal species and associated serovars. After explaining that, you can concentrate on the food supply chain, then look at the impacts on animal and human health.
The introduction section has been reorganized according to the reviewer’s suggestion so it follows a logical sequence. It appears shaded in colour not to cover the changes highlighted in yellow.
- Section 4 ‘Application of predictive microbiology software for Salmonella in foods’ (L282-351) focuses on using predictive modelling software developed by the Authors. However, as a review paper, it may be advisable to examine the current state of research in this area from various perspectives rather than providing a practical illustration and only focus on foods. For example, Machine learning algorithms has been also used for a variety of purposes, such as linking human outbreaks to specific food sources, detecting the occurrence of Salmonella serovars in human infections, and predicting future outbreaks. I encourage you to provide a comprehensive overview of this field and the work that has already been done. Then, if appropriate, you can expose the example already provided.
Thank you very much for this pertinent comment. An extensive review on predictive modelling could be out of the scope of the present review. We agree that showing different examples of predictive modelling software is not a common practice in review papers. However, according to the authors’ opinion, this is a way to illustrate a practical application of the existing knowledge. We believe that inclusion of fit-for-purpose examples would increase the visibility and interest of the paper from a broader audience other than researches (i.e. food operators, health authorities). Nevertheless, this section has been re-arranged as part of section 3 (3.4) with a new title and updated information of novel machine learning algorithms for Salmonella in foods.
- L36-38: The cited reference n.3 is not relevant to justify the two previous statements. Please, use only relevant papers.
This reference has been changed following the reviewer’s suggestion.
- L44: ‘…the increase in the reptile population…’ What do you mean? Owned reptiles? Please, be more specific.
Following the reviewer’s instructions, the word owned has been added to be more specific.
- L45-46: Some rephrasing is needed. Please consider this example: “Additionally, rodents and birds act as Salmonella amplifiers, playing an essential part in the dissemination of bacteria on farms [9].”
Following the reviewer’s instructions, the sentence has been restructured.
- L48-50: Also, here need for rephrasing. Here you find an option: “Salmonella, which can be present in carrier animals' intestinal contents, may cause contamination at various stages of the slaughter process, such as in trucks, lairages, slaughter lines, and quartering [10].”
Following the reviewer’s instructions, the sentence has been restructured.
- L58: “on both bacterium and host characteristics” the bacterium is always Salmonella; I think you mean here the serovar. Please, replace the term. Instead, what do you mean with host characteristics? In terms of immunity? Please, be more factual.
Following the reviewer’s instructions, the missing information has been added to make the sentence understandable.
- L79-83: The concept is repeated here from L59-61.
Following the reviewer’s instructions, the concept has been removed from the second sentence not to be repeated twice.
- Table 1: from the title ‘reported’ is not needed. Please, delete it. The acronym MSs requires clarification. I believe you intend to refer to Member States, so you could include this term in the table title or as a footnote. Additionally, you have rounded up most of the percentages, unlike those from Derby and Coeln serovars. For uniformity, I recommend rounding up these percentages too.
Following the reviewer’s instructions, the word ‘reported’ has been deleted from the Table 1 title, the meaning of MSs has been included in the title too and the % have been rounded up for uniformity.
- L89: the first use of EFSA acronym should be expanded as the European Food Safety Authority (EFSA) Consider the following rephrasing: “Recently, the European Food Safety Authority (EFSA) has reported an increase in the frequency and severity of human infections caused by the Typhimurium serovar and its monophasic variant (mST strains), which are associated with meat products derived from porcine or bovine species.”
Following the reviewer’s instructions, the sentence has been modified.
- L122: support and advice for whom? Collecting data from where?
The European Food Safety Authority (EFSA) collects data from the Member States and makes available to citizens all that information.
- L134: please, move the date of access to the References section
Following the reviewer’s instructions, the information has been moved to the reference section.
- L146: replace ‘puts’ by ‘ranks’
The sentence has been completely rewritten following the suggestion of other reviewer and the word ‘put’ has disappeared.
- L200: They, who? I suggest modifying the text as follows: “… such as NaCl. For instance, Pye et al. [26] compared…”
Following the reviewer’s instructions, the sentence has been restructured.
- L203: ‘while S. Enteritidis showed the highest resistance level for citric acid.’
Following the reviewer’s instructions, the sentence has been restructured.
- L207-211: It is a long sentence. Please, rearrange it, otherwise the meaning of the text can be lost.
The sentence has been re-arranged and summarized in the manuscript.
- L223: ‘and/or’
The sentence has been modified following the reviewer’s suggestion.
- L228: Please start the sentence by stating the author's name instead of just the reference number (e.g., Liu et al. [35]). Additionally, it is advisable to avoid biased language and the use of subjective terms such as 'nice'.
The sentence has been modified following the reviewer’s suggestion.
- L270: Which research? As mentioned above, please provide the name of the author to make explicit what you are referring to.
The sentence has been modified following the reviewer’s suggestion.
- L278: again, what do you mean with ‘the latest research’? the ref. [52]?
We referred to new research trends. This has been corrected.
- L283: What do you mean with ‘microbial fate’?
In this sentence, the term microbial fate refers to Salmonella behaviour over the food chain. This has been changed in the revised manuscript.
- L308-351: This section is a little confusing… The message is sometimes lost. Please, rearrange it.
Thank you very much for your comment. This section has been included as a part of section 3 (3.4) with a new title. This way, we think that it can be better understood.
- L334-336: It should be the opposite; I mean the predictions are relatively close to observations but not the contrary. Please, rearrange the sentence.
Thank you very much for this comment. The sentence has been rearranged.
- L341: Figure 3 in this manuscript version does not correspond with you are mentioning. Maybe did you mean Figure 2?
Corrected.
- Figure 2. Could you provide the Y-axis titles in the middle and bottom graphs?
These graphical representations are automatically generated by the MicroHibro software, so it is not possible to change the axis title. Nevertheless, the figure legend has been clarified including the meaning of x and y axis.
- L376: ‘Kentucky’. Please correct it.
The name has been modified following the reviewer’s suggestion.
- L379: 90,000 cases of resistant salmonellosis? Please follow a logical order in your arguments. You discussed antimicrobial resistance earlier and now you are presenting data on human salmonellosis cases. This has already been addressed at the beginning of the manuscript, so it is unnecessary to include it again here.
The repeated information has been deleted from this section.
- L415-463: References are missing… please provide them.
Following the reviewer’s comment, the reference has been included (EFSA document 2022, reference 3).
- Figure 3 and 4: I am not certain that you can use the same figures as reported by the cited reference. Can you recreate the figure instead of using the same one from the cited reference? You could declare that the modification was made from [67].
According to the Journal Animals guidelines for authors regarding copyrights, there would be no problem on using these figures duly referenced, due to the belong to a public organization (EFSA).
- L511: what do you mean with MIC90 and MBC90. It is the first time they appear in the text. Please, explain it.
The missing information has been added.
- L515: the acronym ‘MEOR’ is not necessary, as you don’t use it anywhere else in the manuscript. Please, provide only the necessary abbreviations throughout the text.
The acronym has been deleted from the sentence, following the reviewer’s suggestion.
- L633-638: Is this still necessary to explain here? It is repetitive from the Introduction section.
The information has been deleted not to be repetitive.
- The reference section should be revised in the format described in the Journal’s guidelines. I have detected some errors in ref. n. 5, 8, 10, 11, 58, 62, 68. Ref. n. 20 needs for a title, the url and the date of access. Ref. n. 22 needs for a title of the consulted section of the website and a date of access.
Following the reviewer’s comment, all the references have been revised and written in the correct format according to journal. The information required has also been added.
- Comments on the Quality of English Language. Although I acknowledge that English is not the authors' native language, I suggest they extensively revise their writing style and, if feasible, employ a more academic language. Additionally, the authors' use of lengthy sentences occasionally makes it challenging to comprehend the intended message, which is unfortunate. Therefore, paraphrasing and replacing terms throughout the text can enhance the arguments' readability and comprehensibility.
Following the reviewer’s comment, the English in the whole manuscript has been deeply revised to improve its quality.
Reviewer 2 Report
Comments and Suggestions for Authors
With respect to the authors' work, the manuscript by Márquez et al. does not contribute enough novel information to be published.
Author Response
We deeply regret this observation, since authors believe that this review offers valuable insights into Salmonella, with a particular focus on its public health implications and control strategies. Given Salmonella's significance as a human health pathogen, we contend that this review serves to provide an updated overview of our current understanding of Salmonella across various domains.
Furthermore, the increasing number of recent studies exploring diverse aspects of Salmonella underscores the need for a comprehensive review of this evolving area. Thus, the primary objective of this review is to collate the most recent research and put them all together, offering a holistic perspective on the current status of Salmonella in public health. This work can be very useful to establish knowledge bases in this field for any author working on this pathogen. We acknowledge that the assessment of scientific novelty can be somewhat subjective. However, based on the aforementioned considerations, we firmly believe that our review stands out due to its substantial contributions to the existing knowledge in this field. We hope that after the changes made following the rest of the reviewer’s suggestions, you will finally consider this work as worthy of being published in the journal Animals.
Reviewer 3 Report
Comments and Suggestions for Authors
As reported by the authors, the manuscript provides an overview of the Salmonella and salmonellosis status. It is a useful update in concepts for health professionals involved in animal health and public health. It is a review that facilitates and improves the understanding of the epidemiology and control of the main indicator pathogen of zoonoses, Salmonella.
The manuscript is very well written. The correct identification of the chapters makes the text easy to understand.
There are only minor annotations:
(i) Line 84: Table 1 is not mentioned in the text.
(ii) Line 148: Insert the meaning of the acronym ECDC.
(iii) Line 190-197: The bibliography is missing.
(iv) In the text, I would insert some food-related regulations about Salmonella spp. (Regulation (EC) No 2073/2005 and Commission Regulation (EC) No 1441/2007 on Microbiological criteria for foodstuffs).
Comments on the Quality of English LanguageMinor editing of English language requires
Author Response
Authors are very grateful for your kind and constructive comments about our original manuscript and your willingness to enrich our work. We hope the questions and comments of the reviewers had been properly addressed and the manuscript will now meet the required standard to be accepted for publication in Animals.
Following, you can find the changes that have been performed, which have been highlighted in the newly revised manuscript:
- Line 84: Table 1 is not mentioned in the text.
Following the reviewer’s comments, this has been solved.
- Line 148: Insert the meaning of the acronym ECDC.
Following the reviewer’s comment, the missing meaning has been added to the text.
- Line 190-197: The bibliography is missing.
The reference of He et al. 2023 has been added in this paragraph.
- In the text, I would insert some food-related regulations about Salmonella spp. (Regulation (EC) No 2073/2005 and Commission Regulation (EC) No 1441/2007 on Microbiological criteria for foodstuffs).
Authors appreciate this valuable suggestion; however, we think that these regulations are not the objective of this work.
Reviewer 4 Report
Comments and Suggestions for Authors
The aim of this review paper is quite up-to-date, as although salmonellosis is currently well controlled and along all the food chain, a One Health approach to know the state of the art of the illness and its spread is of great convenience. Moreover, the inclusion of issues such as antimicrobial resistance are of extreme importance nowadays, as this could be one of the main causes of a worsening in the current situation of salmonellosis. Besides, it could be one of the pinciple reasons of a failure in predictive microbiology approaches. Hence, I consider the issue of the paper of great interest.
Nevertheless, the inclusion of section 4, focused on one specific predictive tool developed by the authors, does not match with a review paper. It could be used as an example only in the context of a bigger frame, when some other tools have been presented and compared, as a sample of a good tool to predict Salmonella behavior. Besides, section 5 lacks of a meaningful explanation about antimicrobial resistances in Salmonella. It only presents data from the EFSA –ECDC report without providing any extra information about the reasons that lay down the development of antimicrobial resistances, cross antimicrobial resistances, etc. In section 6.2.3.1, Salmonella biosecurity programs are not dealt in this section, just some disinfection strategies. Moreover, there are some other issues listed in de attached document.

Author Response
Authors are very grateful for your kind and constructive comments about our original manuscript and your willingness to enrich our work. We hope the questions and comments of the reviewers had been properly addressed and the manuscript will now meet the required standard to be accepted for publication in Animals.
Following, you can find the changes that have been performed, which have been highlighted in the newly revised manuscript:
- The inclusion of section 4, focused on one specific predictive tool developed by the authors, does not match with a review paper. It could be used as an example only in the context of a bigger frame, when some other tools have been presented and compared, as a sample of a good tool to predict Salmonella behavior.
Thank you very much for this pertinent comment. An extensive review on predictive modelling could be out of the scope of the present review. We agree that showing different examples of predictive modelling software is not a common practice in review papers. However, according to the authors’ opinion, this is a way to illustrate a practical application of the existing knowledge. We believe that inclusion of fit-for-purpose examples would increase the visibility and interest of the paper from a broader audience other than researches (i.e. food operators, health authorities). Nevertheless, this section has been re-arranged as part of section 3 (3.4) with a new title and updated information of novel machine learning algorithms for Salmonella in foods.
- Besides, section 5 lacks of a meaningful explanation about antimicrobial resistances in Salmonella. It only presents data from the EFSA –ECDC report without providing any extra information about the reasons that lay down the development of antimicrobial resistances, cross antimicrobial resistances, etc. In section 6.2.3.1, Salmonella biosecurity programs are not dealt in this section, just some disinfection strategies.
The missing information regarding antimicrobial resistance in Salmonella has been added and the section ‘6.2.3.1. Salmonella biosecurity control programs’ is now called ‘6.2.3.1. Salmonella cleaning and disinfection protocols’.
- Line 16: Double space prior to Salmonellosis.
The double space has been deleted.
- Line 56: As you have done with the previous vehicles, some references are required for dairy products, seafood, etc.
According to the reviewer’s suggestion, the bibliography has been added.
- Line 74: Are of gastrointestinal nature.
The text has been changed.
- Line 93: Extra space between paragraphs.95
The extra space has been removed.
- Line 113: identifying.
The word has been corrected.
- Line 128: Consider the removal of the date of consulting the EFSA fact sheet. The same for the reference in line 134. I consider that even the link could be only included in the Reference section. The same in line 303, 353, 381 etc.
All the dates of consulting have been removed from the text and added to references.
- Lines 146-148. Please, rewrite the sentence to make it meaningful.
The sentence has been rewritten.
- Line 152: Furthermore, EFSA data rank it as the leader.
The sentence has been modified.
- Line 156: Why do you write the whole Salmonella denomination after using the abbreviation S. several time before in the same sentence? Section 3.1. More references to this issue might be required. Adaptation of Salmonella to low pH and cross-adaptations have been widely studied and proved by several authors.
Corrected and references added.
- Line 219: Please, write properly the degree symbol.223 The same in subsequent temperatures´ appearance.
This has been corrected in the manuscript.
- Section 3.2: Have you considered including the reasons that lead to an increase of Salmonella resistance in low water activity matrixes? It is interesting to add that it might be linked to the stabilization of membranes, proteins, etc.
A new paragraph has been added about the survival mechanisms of Salmonella against low aw.
- Line 228. 232 Please, include the authors before the reference. The same for line 299.
References have been included.
- Sections 3.1 and 3.2: There are several studies that have proved the cross influence of pH and water activity over thermal resistance of Salmonella. I suggest considering the inclusion of a brief comment about it.
Thank you for your comment. The existence of cross protection mechanisms is mentioned at the beginning of section 3 (first paragraph). Specifically, although there are several studies dealing with the influence of pH and water activity on the thermal resistance of Salmonella in foods, this has been implicitly mentioned in sections 3.1 and 3.2 so that the authors preferred to keep the original text.
- Line 313: Please, consider modifying the denomination “produce” for “vegetable raw products”.
Corrected.
- Line 315: Instead of animal foods, consider including the concept “animal derived foodstuffs”.
Corrected.
- Table 2: Please, minimize it to fit the width with the one of the texts.
Adjusted.
- Line 344: In Figure 2, it is shown…
Corrected.
- Line 398: In the previous line, you have written 26 with numbers, and in this line, 6 with letters. Please, harmonize the way you refer to amounts.
Corrected.
- Line 406: There is a extremely big space empty.
Corrected.
- Section 5. There are no references to the reasons that lead to the increase in antimicrobial resistance in Salmonella. Only data about the EFSA-ECDC report are presented, and they don´t provide any extra info beyond the already presented in the report. To consider its inclusion in the review, these data should be crossed with other in vitro studies referring to this issue, antimicrobial use, sales, etc.
The missing information has been added.
- Line 522. Please, write enrofloxacin without capital letters
The change has been done and the names of antimicrobials have been corrected in the whole document.
- Line 531. Here you could also mention some studies that have investigated the bioaccumulation of essential oils in muscle, or even the possibility of the disruption of the intestinal microbiota.
Some information has been added in this sense in other section.
- Line 545. antimicrobialS
This sentence has been modified following the recommendations of another reviewer.
- Line 616: Please, revise the use of the first and third person. They are mixed.
This sentence has been corrected.
- Line 625: Which guidelines?
The guidelines included in the sanitation programs, which has been clarified in the document.
- Section 6.2.3.1.: Salmonella biosecurity programs are not dealt in this section, just some disinfection strategies. Please, correct this section.
Section corrected.
- Line 672: There is some info included in Conclusions section that does not appear before in the paper. For instance, no mention to the loss of effectiveness of antimicrobial treatment associated to antimicrobial resistance development is previously mentioned on its corresponding section. An explanation of the implications of Salmonella antimicrobial resistance development is required in Section 5.
The missing information has been added and thus the conclusions are now correct.
Round 2
Reviewer 1 Report
Comments and Suggestions for Authors
Dear Authors,
Thank you for addressing most of the comments provided in the previous report. However there is still room for improvement in the manuscript. In particular, I disagree with your opinion that it should be legal to copy and paste something that it is the contribution of other authors or institutions regardless of the lack of copyright or whether it is a public institution. In doing so you are falsely claiming that this is part of your contribution which it is not… As an academic, I ask you, are students allowed to do this in their bachelor’s theses? I don’t think so, and even less so for academic professionals. According to the Journal’s guidelines reporting sentences taken from other reports, papers, etc., should be enclosed by “ ”. As a reviewer, it’s my duty to ensure the scientific integrity and quality of the manuscript. In my opinion, it’s not ethical to do so, you can find other ways to express the same idea/content by using your own words without distorting the official information.
Of note, with such a high percentage (30%) of plagiarism detected, I should have automatically rejected your manuscript, according to the journal’s guidelines. Instead, I and the Editor gave you the opportunity to fix this issue. So, I strongly recommend once again that you edit the compromised text using your own words.
Introduction
There are some parts of the text that still need for ordering. For example, in L52-66 the consequences of NTS infections in humans are explained, and this is somehow repeated in L98-109. I recommend integrating both parts into one paragraph, located in L98-109, to avoid redundancy.
There is also a need for terminology uniformity. Some examples you should address are listed below:
- In L43-44, you referred to “S. Typhimurium and its monophasic variant” without providing the relevant acronym, while in L73-74, you referred to them as “Typhimurium serovar and its monophasic variant (mST strains)”.
- In L74-75, you employed the terms “porcine and bovine species”, then in L86 you use different terminology referring to those species as “swine” and “cattle”
Please, uniform the terms employed throughout the manuscript to ensure cohesivity of your arguments.
L41: I think you mean that the severity depends on the ‘serovar’, because the pathogen is always the same. Please, replace the term.
L42-46: These three sentences should be placed after you explain how the genus is classified or when you focus on NTS in L63-66, thus avoiding redundancy.
L66: ‘as mentioned in (Table 1)’. Please replace ‘mentioned’ by ‘illustrated’ and report ‘Table 1’ without brackets.
L76-84: The paragraph lacks coherence and is excessively verbose. It requires rearrangement and should be merged with the paragraph in L110-115 to maintain continuity between the paragraphs. Moreover, you are outlining here the methods used for identifying Salmonella, however, a comparison between them is missing. Therefore, the phrase 'On the other hand' is not suitable in this context. Please, consider the following paraphrasing: “Salmonella strain typing is a crucial component of routine laboratory investigations [8]. Phage typing and serotyping, as well as molecular methods, are essential tools for this purpose. It enables the identification and isolation of such strains from primary animal sources as well as non-animal sources (i.e., food, water, and environmental samples). The most commonly utilised methods include pulsing-field gel electrophoresis (PFGE) and multiple locus variable analysis (MLVA). New genome-based typing methods, such as whole genome sequencing (WGS), are employed to track outbreaks and determine the epidemiological origin of the infection [9,10,11].”
L92: ‘Salmonella’ should be in italics. Check and correct it when necessary thoughout the manuscript (L198, 222, etc.)
L98-105: This fragment is completely the same paragraph as reported in the third paragraph of the introduction section of the following publication (https://doi.org/10.3390/microorganisms11071765). Please, re-write it.
Section 2
L152: ‘behind Campylobacter’. The correct form should be ‘after Campylobacter’.
L154: ‘According to the European Centre of Disease Prevention and Control (ECDC) data’. This is not necessary, since you are referring to the zoonoses report mentioned in the previous sentence. Please, start the sentence immediately with “The number of confirmed cases…”
L162-173: This fragment is completely the same paragraph as reported in the seventh paragraph of the introduction section of the following publication (https://doi.org/10.3390/microorganisms11071765). Please, re-write it.
Section 3
L216-218: The two sentences to express the same concept, please delete at your convenience.
L220-221: ‘potentially informing’ replace it by ‘which may lead to’
L223-226: Please, consider the following paraphrasing: “The high frequency of Salmonella in low-water activity (aw) foods (such as powders, flours, dried fruits, spices, oily foods, and nuts) is a cause for concern. Recent studies have extensively reported this situation due to the growing number of salmonellosis outbreaks related to these products.”
L235-239: Consider to paraphrase this part of the text for a better comprehension: “Adaptive responses in Salmonella help it survive by accumulating compatible solutes including proline, glycine, betaine, ectoine, and trehalose, leading to reduced water loss [36]. Furthermore, osmoregulation plays a vital role in maintaining the turgor pressure of the bacteria through increasing the intracellular concentration of compatible solutes [37,38].”
L297-395: I keep thinking that the section dedicated to predictive model requires improvement for a review paper. fAn assessment of various models is necessary, rather than focusing sololy on the Authors' developed model, which is unfit for this context.
Section 4
L398-422: This fragment is highly similar to the contents reported in the following publication: Annous, B. (Ed.). (2012). Salmonella - Distribution, Adaptation, Control Measures and Molecular Technologies. InTech. doi: 10.5772/2470
L404-409: This fragment is highly similar to the contents reported in the following publication: https://doi.org/10.1016/j.rvsc.2022.09.028
L414-417: This fragment is completely the same paragraph as reported in the following publication: https://efsa.onlinelibrary.wiley.com/doi/10.2903/j.efsa.2020.6007
L423-429: This fragment is completely the same paragraph as reported in the conclusions section of the following publication (https://doi.org/10.3390/microorganisms9102018). Please, re-write it.
L440-445: This fragment is completely the same paragraph as reported in the abstract and part of the conclusions sections of the following publication (https://doi.org/10.3390/microorganisms9102018). Please, re-write it.
L446-521: The complete Section 4.2. Key findings must be re-written.
L473-477: This fragment is completely the same paragraph as reported in the abstract of the following publication (https://doi.org/10.3390/microorganisms11071765). Please, re-write it.
Figure 3 and 4 are unnecessary, since they are obtained from another document.
L722-725: This fragment is completely the same paragraph as reported in the conclusions section of the following publication (https://doi.org/10.3390/microorganisms9102018). Please, re-write it.
Comments on the Quality of English Language
The quality of English language has been improved, some minor editing is still needed.
Author Response
Introduction
There are some parts of the text that still need for ordering. For example, in L52-66 the consequences of NTS infections in humans are explained, and this is somehow repeated in L98-109. I recommend integrating both parts into one paragraph, located in L98-109, to avoid redundancy.
Corrected
There is also a need for terminology uniformity. Some examples you should address are listed below:
- In L43-44, you referred to “S. Typhimurium and its monophasic variant” without providing the relevant acronym, while in L73-74, you referred to them as “Typhimurium serovar and its monophasic variant (mST strains)”.
Uniformed S. Typhimurium and its monophasic variant
- In L74-75, you employed the terms “porcine and bovine species”, then in L86 you use different terminology referring to those species as “swine” and “cattle”
Uniformed
L41: I think you mean that the severity depends on the ‘serovar’, because the pathogen is always the same. Please, replace the term.
Replaced
L42-46: These three sentences should be placed after you explain how the genus is classified or when you focus on NTS in L63-66, thus avoiding redundancy.
Replaced
L66: ‘as mentioned in (Table 1)’. Please replace ‘mentioned’ by ‘illustrated’ and report ‘Table 1’ without brackets.
Replaced
L76-84: The paragraph lacks coherence and is excessively verbose. It requires rearrangement and should be merged with the paragraph in L110-115 to maintain continuity between the paragraphs. Moreover, you are outlining here the methods used for identifying Salmonella, however, a comparison between them is missing. Therefore, the phrase 'On the other hand' is not suitable in this context. Please, consider the following paraphrasing: “Salmonella strain typing is a crucial component of routine laboratory investigations [8]. Phage typing and serotyping, as well as molecular methods, are essential tools for this purpose. It enables the identification and isolation of such strains from primary animal sources as well as non-animal sources (i.e., food, water, and environmental samples). The most commonly utilised methods include pulsing-field gel electrophoresis (PFGE) and multiple locus variable analysis (MLVA). New genome-based typing methods, such as whole genome sequencing (WGS), are employed to track outbreaks and determine the epidemiological origin of the infection [9,10,11].”
Modified
L92: ‘Salmonella’ should be in italics. Check and correct it when necessary thoughout the manuscript (L198, 222, etc.)
Corrected
L98-105: This fragment is completely the same paragraph as reported in the third paragraph of the introduction section of the following publication (https://doi.org/10.3390/microorganisms11071765). Please, re-write it.
Rewritten
Section 2
L152: ‘behind Campylobacter’. The correct form should be ‘after Campylobacter’.
Corrected
L154: ‘According to the European Centre of Disease Prevention and Control (ECDC) data’. This is not necessary, since you are referring to the zoonoses report mentioned in the previous sentence. Please, start the sentence immediately with “The number of confirmed cases…”
Corrected
L162-173: This fragment is completely the same paragraph as reported in the seventh paragraph of the introduction section of the following publication (https://doi.org/10.3390/microorganisms11071765). Please, re-write it.
Rewritten
Section 3
L216-218: The two sentences to express the same concept, please delete at your convenience.
Deleted
L220-221: ‘potentially informing’ replace it by ‘which may lead to’
Replaced
L223-226: Please, consider the following paraphrasing: “The high frequency of Salmonella in low-water activity (aw) foods (such as powders, flours, dried fruits, spices, oily foods, and nuts) is a cause for concern. Recent studies have extensively reported this situation due to the growing number of salmonellosis outbreaks related to these products.”
Changed
L235-239: Consider to paraphrase this part of the text for a better comprehension: “Adaptive responses in Salmonella help it survive by accumulating compatible solutes including proline, glycine, betaine, ectoine, and trehalose, leading to reduced water loss [36]. Furthermore, osmoregulation plays a vital role in maintaining the turgor pressure of the bacteria through increasing the intracellular concentration of compatible solutes [37,38].”
Changed
L297-395: I keep thinking that the section dedicated to predictive model requires improvement for a review paper. fAn assessment of various models is necessary, rather than focusing sololy on the Authors' developed model, which is unfit for this context.
Sección 3.4. has been modified
Section 4
L398-422: This fragment is highly similar to the contents reported in the following publication: Annous, B. (Ed.). (2012). Salmonella - Distribution, Adaptation, Control Measures and Molecular Technologies. InTech. doi: 10.5772/2470
Rewritten
L404-409: This fragment is highly similar to the contents reported in the following publication: https://doi.org/10.1016/j.rvsc.2022.09.028
Rewritten
L414-417: This fragment is completely the same paragraph as reported in the following publication: https://efsa.onlinelibrary.wiley.com/doi/10.2903/j.efsa.2020.6007
Rewritten
L423-429: This fragment is completely the same paragraph as reported in the conclusions section of the following publication (https://doi.org/10.3390/microorganisms9102018). Please, re-write it.
Rewritten
L440-445: This fragment is completely the same paragraph as reported in the abstract and part of the conclusions sections of the following publication (https://doi.org/10.3390/microorganisms9102018). Please, re-write it.
Rewritten
L446-521: The complete Section 4.2. Key findings must be re-written.
Rewritten according to reviewer’s suggestion.
L473-477: This fragment is completely the same paragraph as reported in the abstract of the following publication (https://doi.org/10.3390/microorganisms11071765). Please, re-write it.
Figure 3 and 4 are unnecessary, since they are obtained from another document.
They have been deleted.
L722-725: This fragment is completely the same paragraph as reported in the conclusions section of the following publication (https://doi.org/10.3390/microorganisms9102018). Please, re-write it.
Rewritten
Reviewer 2 Report
Comments and Suggestions for Authors
The new version of the manuscript entitled “Salmonella and salmonellosis: an update on public health implications and control strategies” by Márquez et al. has been greatly improved in terms of structure and data provided, compared to the previous draft. However, this study requires some improvements, which are highlighted below and which will allow the publication in the prestigious journal Animals.
Please explain the selection strategy for the articles considered in the scientific manuscript.
In subchapter 3.3. The biofilm formation of Salmonella in food processing environments, the authors should extend the information by considering a valuable recently published article (doi: 10.3390/life12101618).
In chapter 4. Antimicrobial resistance the authors should expand the discussion on the resistance of Salmonella to biocides used in the food industry. In this regard the authors should consult a scientifically sound review article (doi: 10.1016/j.foodres.2011.02.002).
Author Response
The new version of the manuscript entitled “Salmonella and salmonellosis: an update on public health implications and control strategies” by Márquez et al. has been greatly improved in terms of structure and data provided, compared to the previous draft. However, this study requires some improvements, which are highlighted below and which will allow the publication in the prestigious journal Animals.
Please explain the selection strategy for the articles considered in the scientific manuscript.
The selection of the bibliography has been carried out based on the updates available in each of the topics included in the review.
In subchapter 3.3. The biofilm formation of Salmonella in food processing environments, the authors should extend the information by considering a valuable recently published article (doi: 10.3390/life12101618).
Information extended
In chapter 4. Antimicrobial resistance the authors should expand the discussion on the resistance of Salmonella to biocides used in the food industry. In this regard the authors should consult a scientifically sound review article (doi: 10.1016/j.foodres.2011.02.002).
Included
Reviewer 4 Report
Comments and Suggestions for Authors
Thanks for yor attempt on improving the paper. Nevertheless, it lacks novelty. Most of the data presented are included on official reports. Moreover, the inclusion of the predictive model does not fit on a review paper. It should include a comparison between several models, but centering on one model developed by the authors has no point on this context. Besides, the inclusion of graphs obtained from another document (Figures 3 and 4, again from an official report) is unncessary.
Author Response
Thanks for your attempt on improving the paper. Nevertheless, it lacks novelty. Most of the data presented are included on official reports. Moreover, the inclusion of the predictive model does not fit on a review paper. It should include a comparison between several models, but centering on one model developed by the authors has no point on this context. Besides, the inclusion of graphs obtained from another document (Figures 3 and 4, again from an official report) is unnecessary.
Figures 3 and 4 have been eliminated.
Round 3
Reviewer 1 Report
Comments and Suggestions for Authors
Dear Authors,
thank you for having addressed my suggestions.
I accept the manuscript in the present form.
Kind regards.
Reviewer 2 Report
Comments and Suggestions for Authors
The latest version of the manuscript entitled “Salmonella and salmonellosis: an update on public health implications and control strategies” by Márquez et al. has been improved according to the proposed suggestions.
Thank you for considering all my recommendations!